# Consistency Models as Plug-and-Play Priors for Inverse Problems

## Abstract

Diffusion models have found extensive use in solving numerous inverse problems. Such diffusion inverse problem solvers aim to sample from the posterior distribution of data given the measurements, using a combination of the unconditional score function and an approximation of the posterior related to the forward process. Recently, consistency models (CMs) have been proposed to directly predict the final output from any point on the diffusion ODE trajectory, enabling high-quality sampling in just a few NFEs. CMs have also been utilized for inverse problems, but existing CM-based solvers either require additional task-specific training or utilize data fidelity operations with slow convergence, not amenable to large-scale problems. In this work, we reinterpret CMs as proximal operators of a prior, enabling their integration into plug-and-play (PnP) frameworks. We propose a solver based on PnP-ADMM, which enables us to leverage the fast convergence of conjugate gradient method. We further accelerate this with noise injection and momentum, dubbed **PnP-CM**, and show it maintains the convergence properties of the baseline PnP-ADMM. We evaluate our approach on a variety of inverse problems, including inpainting, super-resolution, Gaussian and nonlinear deblurring, and magnetic resonance imaging (MRI) reconstruction. To the best of our knowledge, this is the *first CM trained for MRI* datasets. Our results show that PnP-CM achieves high-quality reconstructions in as few as 4 NFEs, and can produce meaningful results in 2 steps, highlighting its effectiveness in real-world inverse problems while outperforming comparable CM-based approaches.

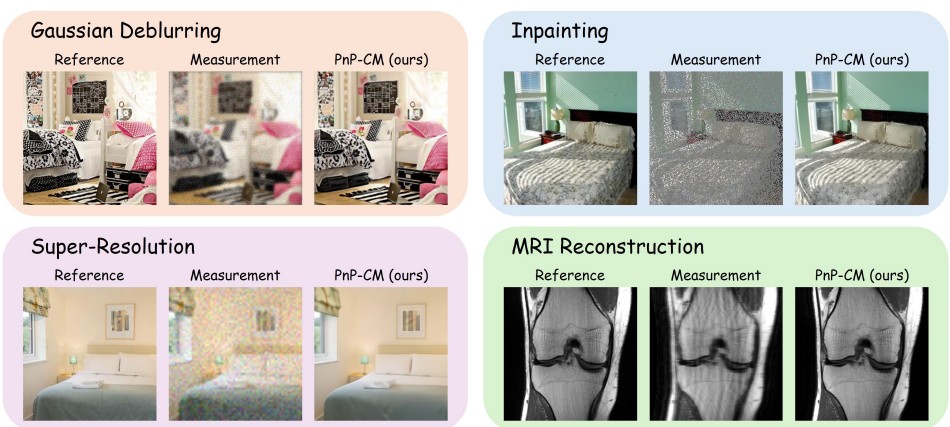

Figure 1: Representative results of our method (PnP-CM) on four distinct noisy inverse problems: Gaussian deblurring, inpainting, and super-resolution (all with Gaussian noise $\sigma = 0.05$), as well as MRI reconstruction with inherently noisy measurements.

## 1 Introduction

Diffusion models (DMs) have established themselves as state-of-the-art generative models, capable of synthesizing high-fidelity and diverse samples across a wide range of domains (Ho et al., 2020; Song et al., 2021a;b; Dhariwal & Nichol, 2021; Karras et al., 2022). Beyond unconditional generation, DMs have also been widely adopted as powerful priors for solving imaging inverse

problems (Chung et al., 2023a; Song et al., 2023a; Wang et al., 2023), where the goal is to recover clean images from degraded measurements. Conventional DM-based solvers typically combine the unconditional score function with a data-consistency term tied to the forward operator, and generate reconstructions by integrating either the reverse stochastic differential equation (SDE) or the probability flow ordinary differential equation (ODE) (Song et al., 2021b). While highly effective, these approaches are computationally expensive, often requiring hundreds of neural function evaluations (NFEs) to achieve competitive results, which limits their practicality in large-scale or time-sensitive applications (Salimans & Ho, 2022; Song et al., 2023b).

Several approaches have been proposed to accelerate DM sampling, including progressive distillation (Salimans & Ho, 2022; Meng et al., 2023), rectified flow (Liu et al., 2023b; 2024), and diffusion matching distillation (DMDs) (Yin et al., 2024b;a). Among these, consistency models (CMs) (Song et al., 2023b; Lu & Song, 2025) distill the generative power of DMs into a consistency function that maps any point on the diffusion ODE trajectory directly to its clean origin. This design enables high-quality sampling in as few as one to four NFEs, which has made CMs particularly attractive as efficient priors for inverse problems. However, existing CM-based methods face limitations. For instance, CoSIGN (Zhao et al., 2024) incorporates an external ControlNet (Zhang et al., 2023) to encode measurement operators, requiring task-specific training for each degradation. This restricts its generalization across inverse problems, since adapting to a new measurement operator typically entails retraining or fine-tuning the ControlNet. Similarly, CM4IR (Garber & Tirer, 2025) leverages a form of pseudo-inverse guidance for measurement consistency, a strategy fundamentally tied to the conditioning of the forward operator. Thus, its use in highly ill-conditioned systems is nontrivial, as the pseudo-inverse becomes unstable, thereby limiting its reliability in more complex imaging tasks.

In this work, we propose **PnP-CM**, a novel framework that reinterprets consistency models (CMs) as proximal operators within a learned prior setting. This perspective allows CMs to be seamlessly embedded into plug-and-play (PnP) optimization (Venkatakrishnan et al., 2013), combining their sampling efficiency with the convergence guarantees of PnP methods. Our solver builds on the alternating direction method of multipliers (ADMM), with enhancements to speed-up convergence while preserving theoretical guarantees. Representative results of our method are shown in Fig. 1. Our main contributions are as follows:

- We introduce **PnP-CM**, a PnP-ADMM solver that reinterprets consistency models (CMs) as proximal operators of a prior. This allows the use of conjugate gradient methods for data fidelity enabling fewer outer iterations. We further enhance the runtime of PnP-CM using Nesterov momentum, and controlled noise injection, while showing that they maintain the convergence properties of the baseline algorithm.
- We evaluate PnP-CM on LSUN Bedroom (Yu et al., 2015) and fastMRI (Knoll et al., 2020) for inpainting, super-resolution, Gaussian and nonlinear deblurring, and magnetic resonance imaging (MRI) reconstruction. To the best of our knowledge, this is the first work to train a consistency model directly for MRI.
- PnP-CM achieves state-of-the-art results in as few as four NFEs and produces meaningful outputs in just two steps, consistently outperforming comparison baselines in both quality and efficiency.

## 2 BACKGROUND AND RELATED WORK

### 2.1 DIFFUSION MODELS

DMs (Ho et al., 2020; Song et al., 2021b; Karras et al., 2022) define a generative process by progressively corrupting data with Gaussian noise and then learning how to invert this process. Let $p_{\text{data}}(\mathbf{x})$ denote the data distribution. The forward noising process is often described by the stochastic differential equation (SDE) (Song et al., 2021b)

$$d\mathbf{x}_t = \mu(\mathbf{x}_t, t)\, dt + \sigma(t)\, dw_t, \quad t \in [0, T], \tag{1}$$

where $w_t$ is standard Brownian motion, and $\mu$ and $\sigma$ specify the drift and diffusion schedules. This ensures that the marginal distribution of $\mathbf{x}_t$, denoted $p_t(\mathbf{x})$, evolves smoothly from $p_0(\mathbf{x}) = p_{\text{data}}(\mathbf{x})$ to a tractable prior $p_T(\mathbf{x})$, typically close to a Gaussian distribution. An important observation is that the SDE in Eq. 1 is associated with a deterministic probability flow ODE (Song et al., 2021b),

$$d\mathbf{x}_t = \left[ \mu(\mathbf{x}_t, t) - \tfrac{1}{2}\sigma^2(t)\, \nabla \log p_t(\mathbf{x}_t) \right] dt, \tag{2}$$

which shares the same marginal distributions as the SDE. Because the drift term involves the score function $\nabla \log p_t(\mathbf{x})$, diffusion models are often viewed as score-based generative models. In practice, one learns a neural score network $s_\phi(\mathbf{x}, t)$ to approximate $\nabla \log p_t(\mathbf{x})$ using denoising score matching. Substituting this estimate into Eq. 2 yields the empirical probability flow ODE:

$$\frac{\mathrm{d}\mathbf{x}_t}{\mathrm{d}t} = -t\, s_\phi(\mathbf{x}_t, t). \tag{3}$$

To generate new samples, one initializes from the prior distribution $\mathbf{x}_T \sim \pi(\mathbf{x}) \approx \mathcal{N}(0, T^2 I)$ and integrates Eq. 3 backward in time to $t = 0$ using a numerical solver such as Euler (Song et al., 2021a;b) or Heun's (Karras et al., 2022) method. The terminal point $\mathbf{x}_0$ is then regarded as a sample from $p_{\text{data}}(\mathbf{x})$. For stability, the integration is typically stopped at a small $\varepsilon > 0$ instead of exactly at zero.

## 2.2 CONSISTENCY MODELS

Consistency models (CMs) (Song et al., 2023b; Lu & Song, 2025) were proposed to accelerate diffusion sampling while preserving sample quality. Instead of sequentially denoising through the entire diffusion trajectory, CMs learn a single mapping that directly projects a noisy sample $\mathbf{x}_t$ at any timestep $t$ to its corresponding clean origin $\mathbf{x}_0$. Formally, a consistency function $f_\theta(\mathbf{x}_t, t)$ is trained to satisfy

$$f_\theta(\mathbf{x}_t, t) = f_\theta(\mathbf{x}_{t'}, t'), \quad \forall t, t' \in [\epsilon, T], \tag{4}$$

with the boundary condition $f_\theta(\mathbf{x}_\epsilon, \epsilon) = \mathbf{x}_\epsilon$. Here $\mathbf{x}_t$ and $\mathbf{x}_{t'}$ are drawn from the forward process defined in Eq. 1. Training can be carried out either via consistency distillation (CD) by distilling the behavior of an existing DM or by consistency training (CT) which learns $f_\theta$ directly from data. In both cases, the objective enforces that predictions from different noisy inputs of the same target agree, leading to the consistency loss

$$\mathcal{L}_{\text{CM}} = \mathbb{E}_{\mathbf{x}_0, t, \mathbf{x}_t, t', \mathbf{x}_{t'}} \left[ w(t) d(f_\theta(\mathbf{x}_t, t), f_{\theta^-}(\mathbf{x}_{t'}, t')) \right], \tag{5}$$

where $(\mathbf{x}_t, t)$ and $(\mathbf{x}_{t'}, t')$ are two independently sampled noised versions of $\mathbf{x}_0$ at timesteps $t$ and $t'$, $w(t)$ is a weighting function, $d(\cdot, \cdot)$ is a distance function such as $\ell_2$ or LPIPS, and $f_{\theta^-}$ describes a corresponding "teacher" network achieved by exponential moving average (EMA). At inference, sampling does not need to begin from pure Gaussian noise. Instead, CMs allow initialization from any intermediate state $\mathbf{x}_t$, and the consistency function maps it directly toward the clean estimate:

$$\hat{\mathbf{x}}_0 = f_\theta(\mathbf{x}_t, t), \quad t \in (0, T]. \tag{6}$$

In practice, one may start from $\mathbf{x}_T \sim \pi(\mathbf{x})$, where $\pi(\mathbf{x})$ denotes the prior distribution at time $T$ or from a measurement-dependent initialization, and apply $f_\theta$ either once or over a few discretized steps for improved stability. This flexibility enables high-quality image generation in as few as 1–4 NFEs, in stark contrast to the hundreds typically required by diffusion-based methods.

## 2.3 GENERATIVE APPROACHES TO INVERSE PROBLEMS

Inverse problems aim to recover an unknown signal $\mathbf{x} \in \mathbb{R}^n$ from a set of degraded observations $\mathbf{y} \in \mathbb{R}^m$. The measurement process is typically modeled as

$$\mathbf{y} = \mathbf{A}\mathbf{x} + \mathbf{n}, \tag{7}$$

where $\mathbf{A} \in \mathbb{R}^{m \times n}$ is the forward operator, and $\mathbf{n}$ is measurement noise. For $m < n$, the problem is ill-posed, and a common strategy to make it tractable is maximum a posteriori (MAP) estimation:

$$\hat{\mathbf{x}} = \arg\max_{\mathbf{x}} p(\mathbf{x} \mid \mathbf{y}) = \arg\min_{\mathbf{x}} -\log p(\mathbf{y} \mid \mathbf{x}) - \log p(\mathbf{x}), \tag{8}$$

where the likelihood term $-\log p(\mathbf{y} \mid \mathbf{x})$ is the *data fidelity* and the prior term $-\log p(\mathbf{x})$ enforces natural image statistics.

**DM-based approaches.** These adapt generative DMS to approximate the posterior distribution $p(\mathbf{x}|\mathbf{y})$ by combining score-based priors with data-consistency terms. A pioneering example is diffusion posterior sampling (DPS) (Chung et al., 2023a), which modifies the denoising step by incorporating the gradient with respect to $\mathbf{x}_t$ of the measurement likelihood evaluated at the denoised Tweedie estimate of the clean data, as an approximation to $\log p(\mathbf{y}|\mathbf{x}_t)$. Intuitively, this adjustment

encourages each denoising update to remain consistent with the observed measurements while following the learned data prior. While effective, DPS typically requires hundreds of NFEs to achieve stable reconstructions, which severely limits its practicality. More recent approaches (Song et al., 2023a; Zhu et al., 2023; Mardani et al., 2024; Alçalar & Akçakaya, 2024; Moufad et al., 2025) have reduced the computational cost to roughly 20–100 NFEs, but this is still far from the efficiency required for large-scale or real-time applications. A complementary line of work formulates inverse problems as Schrödinger bridge problems, interpolating between natural image priors and measurement-conditioned distributions through coupled forward–reverse dynamics. Approaches such as I$^2$SB (Liu et al., 2023a) and CDDB (Chung et al., 2023b) achieve faster sampling by initializing from measurements rather than Gaussian noise, but since the generative model must be trained from scratch for each forward operator, they generalize poorly to new or out-of-domain tasks.

**CM-based solvers.** CMs (Song et al., 2023b; Lu & Song, 2025) offer another direction by distilling diffusion trajectories into a direct mapping from noisy states to clean reconstructions, enabling sampling in as few as 1–4 NFEs. CoSIGN (Zhao et al., 2024) extends this framework by integrating a ControlNet to encode forward operators, but requires retraining or fine-tuning for each new measurement model. CM4IR (Garber & Tirer, 2025) instead combines CMs with back-projection using a pseudo-inverse to enforce measurement consistency, a formulation that is effective for linear operators but does not readily extend to nonlinear or ill-conditioned cases. These limitations underscore the need for CM-based methods that are general and efficient.

## 3 PNP-CM: A CONSISTENCY MODEL INVERSE PROBLEM SOLVER VIA PLUG-AND-PLAY ADMM

In this work, we propose a PnP-ADMM-based inverse problem solver that reinterprets consistency models (CMs) as a proximal operator of a prior. To restore images in only a few NFEs, we further accelerate the method using noise injection and Nesterov momentum (Nesterov, 1983), while preserving convergence guarantees.

### 3.1 USING CMS IN PNP-ADMM

An alternative optimization-centric way to view the MAP estimation of Eq. 8 is:

$$\arg \min_{\mathbf{x}} f(\mathbf{x}) + \lambda g(\mathbf{x}), \tag{9}$$

where $f(\cdot)$ is a data fidelity term, while $g(\cdot)$ is a regularizer/prior (Tibshirani, 1996; Wipf & Rao, 2004; Bishop & Nasrabadi, 2006; Park & Casella, 2008). In particular, for additive i.i.d. Gaussian noise, commonly encountered in computational imaging problems (Chung et al., 2023a; Hammernik et al., 2023), $f(x) = \frac{1}{2}\|\mathbf{y} - \mathbf{A}\mathbf{x}\|_2^2$. ADMM solves (9) by variable splitting with an augmented Lagrangian penalty, and alternating minimization:

$$\mathbf{z}^{(k+1)} = \arg \min_{\mathbf{z}} \ f(\mathbf{z}) + \frac{\rho}{2}\|\mathbf{z} - \mathbf{x}^{(k)} + \mathbf{u}^{(k)}\|_2^2, \tag{10}$$

$$\mathbf{x}^{(k+1)} = \arg \min_{\mathbf{x}} \ g(\mathbf{x}) + \frac{\rho}{2}\|\mathbf{z}^{(k+1)} - \mathbf{x} + \mathbf{u}^{(k)}\|_2^2, \tag{11}$$

$$\mathbf{u}^{(k+1)} = \mathbf{u}^{(k)} + \mathbf{z}^{(k+1)} - \mathbf{x}^{(k+1)}, \tag{12}$$

where the $\mathbf{z}$-update enforces data fidelity, the $\mathbf{x}$-update applies the proximal operator associated with the prior, and $\mathbf{u}$ is the dual variable (Boyd et al., 2011; Hong et al., 2016; Eckstein & Yao, 2015). In practice, the proximal operator of $g(\cdot)$ is often intractable. Plug-and-Play (PnP) methods (Venkatakrishnan et al., 2013; Kamilov et al., 2017; Chan et al., 2016) overcome this by replacing the proximal update with an off-the-shelf denoiser $D_\sigma(\cdot)$:

$$\mathbf{x}^{(k+1)} = D_{\sigma_k}\left(\mathbf{z}^{(k+1)} + \mathbf{u}^{(k)}\right) \tag{13}$$

Such denoisers can also be based on deep neural networks, thus allowing to utilize ability to learn rich image priors for improved restoration quality (Chan et al., 2016). Thus, PnP-ADMM provides a framework that integrates deep denoisers, enabling stable and high-quality reconstructions across a wide range of inverse problems (Chan, 2019).

One of the other advantages of the ADMM baseline is related to the quadratic penalty in Eq. 10. This effectively results in Tikhonov regularization for the data fidelity, improving the conditioning of the minimization sub-problem, making it less sensitive to the conditioning of the forward operator compared to methods based on proximal gradient descent (PGD) (Deng & Yin, 2016; Hong & Luo, 2017). In fact, existing CM-based methods, such as CM4IR (Garber & Tirer, 2025) can be seen as a type of preconditioned PGD with data fidelity based on a backprojection objective, $\frac{1}{2}||\mathbf{A}^{\dagger}(\mathbf{y} - \mathbf{A}\mathbf{x})||_2^2$ (Tirer & Giryes, 2020), which inherits the dependencies of the baseline algorithm on the conditioning of the forward operator. Furthermore, it is well-understood that ADMM requires fewer iterations compared to PGD in practice for convergence to modest accuracy, sufficient for imaging inverse problems (Boyd et al., 2011). Finally, for large-scale systems, on which singular value decomposition is challenging, Eq. 10 can be solved with the conjugate gradient (CG) method, whose speed-up benefits over gradient descent-type approaches has also been noted in the context of DM-based inverse problem solvers (Chung et al., 2024).

In this study, we use the CMs as the denoisers in Eq. 13. However, the use of CMs alone in the PnP-ADMM framework, even with all its inherent advantages, does not lead to desirable quality in a few NFEs. Thus, to improve performance with limited outer iterations, we propose enhancements based on noise injection and Nesterov momentum to accelerate the algorithm, as described next.

## 3.2 ENHANCEMENTS FOR FASTER PNP-CM

Since our goal is to perform CM-based reconstruction in as few as 2 NFEs, we take additional steps to accelerate ADMM. The aim is to ensure a good solution is reached within a few outer iterations of PnP-CM, while also maintaining any convergence properties of the baseline PnP-ADMM if run until convergence.

**Noise Injection.** A common strategy in CMs is noise injection to improve generative quality (Song et al., 2023b; Song & Dhariwal, 2024; Lu & Song, 2025; Garber & Tirer, 2025). This idea has also been explored in the context of image restoration and inverse problems for a longer time (Egiazarian et al., 2007; Atchadé et al., 2017). In broader optimization community, noise injection was shown to help iterative optimization methods escape saddle points and explore the solution space more effectively (Huang, 2021; Guo et al., 2020). We build on the ideas from these works, and add controlled perturbations for the CM to operate at higher noise levels, improving its performance in the limited iteration regime, with the following guarantee:

**Theorem 1.** *Consider the PnP-ADMM algorithm, where the proximal operator is replaced by an $L$-Lipschitz continuous denoiser $D_\sigma$. Suppose noise injection is applied to the denoiser input,* i.e.

$$\mathbf{x}^{(k+1)} = D_\sigma(\mathbf{z}^{(k+1)} + \mathbf{u}^{(k)} + \boldsymbol{\eta}_k)$$

*where $\mathbf{x}$ and $\mathbf{z}$ are the primal variables, $\mathbf{u}$ is the dual variable, $\boldsymbol{\eta}$ is an injected noise term and $k$ counts the iterations. If the PnP-ADMM algorithm without noise injection converges to a fixed point, then the algorithm with noise injection still converges, provided the noise sequence $\{\boldsymbol{\eta}_k\}$ is diminishing and satisfies $\sum_{k=0}^{\infty} ||\boldsymbol{\eta}_k||_2 < \infty$.*

This result states that noise injection does not compromise the existing convergence properties of ADMM, if the noise amplitudes are chosen to satisfy an energy bound. The proof is provided in Appendix A. We note that in addition to the theoretical guarantees we provided, our approach has a practical difference to noise injection in existing CM-based inverse problem solvers (Garber & Tirer, 2025), which uses a correction term based on the previous noise instance. Our approach, on the other hand, generates the noise in a random manner, consistent with previous works from inverse problems (Egiazarian et al., 2007; Atchadé et al., 2017).

**Nesterov Momentum.** To accelerate convergence and reduce the number of neural function evaluations (NFEs), we augment the ADMM updates with a Nesterov-type momentum term. While momentum is well known to improve convergence speed (Nesterov, 1983; Thorley et al., 2021), it can also introduce the risk of divergence if applied indiscriminately (Assran & Rabbat, 2020; Wang et al., 2022). Several algorithms have been proposed to mitigate this risk, such as restart schemes that monitor the progress of the iterates and reset the momentum when divergence is detected (Goldstein et al., 2014; Su et al., 2016). These methods provide theoretical guarantees by ensuring that momentum does not degrade convergence. Drawing from these insights, we conclude that with

---

**Algorithm 1** PnP-ADMM with Consistency Models (PnP-CM)

---

**Require:** Consistency model $f_\theta(\cdot, t)$, sequence of time points $\{t_n\}_{n=0}^N$, penalty parameters $\{\rho_n\}_{n=1}^N > 0$, momentum coefficients $\{\beta_n\}_{n=1}^N > 0$, forward operator $\mathbf{A}$, measurement $\mathbf{y}$
 1: **Initialize:** $\hat{\mathbf{x}}_N \leftarrow 0, \ \hat{\mathbf{u}}_N \leftarrow 0, \ \mathbf{x}_N \leftarrow 0, \ \mathbf{u}_N \leftarrow 0$
 2: **for** $n$ from $N-1$ to $0$ **do**
 3:     $\mathbf{z}_n = (\mathbf{A}^\top \mathbf{A} + \rho_{n+1}\mathbf{I})^{-1}(\mathbf{A}^\top \mathbf{y} + \rho_{n+1}(\hat{\mathbf{x}}_{n+1} - \hat{\mathbf{u}}_{n+1}))$         # Data-fidelity update
 4:     $\boldsymbol{\nu}_n \sim \mathcal{N}(\mathbf{z}_n + \hat{\mathbf{u}}_{n+1}, t_{n+1}^2\mathbf{I})$         # Noise injection
 5:     $\mathbf{x}_n = f_\theta(\boldsymbol{\nu}_n, t_{n+1})$         # CM proximal operator
 6:     $\mathbf{u}_n = \hat{\mathbf{u}}_{n+1} + \mathbf{z}_n - \mathbf{x}_n$         # Dual variable update
 7:     $\hat{\mathbf{x}}_n = \mathbf{x}_n + \beta_{n+1}(\mathbf{x}_n - \mathbf{x}_{n+1})$         # Primal momentum update
 8:     $\hat{\mathbf{u}}_n = \mathbf{u}_n + \beta_{n+1}(\mathbf{u}_n - \mathbf{u}_{n+1})$         # Dual momentum update
 9: **end for**
10: **Return:** $\mathbf{x}_0$

---

carefully chosen momentum coefficients, divergence can be avoided while still benefiting from the acceleration effect. In our limited iteration setup, the empirical acceleration from momentum is achieved by fine-tuning the associated coefficients heuristically.

**Final Algorithm.** The complete algorithm is outlined in Alg. 1. As detailed, our approach combines ADMM-PnP with noise injection and Nesterov momentum to improve output quality with fewer iterations, while preserving convergence properties under suitable conditions. Note that, to maintain consistency with common DM and CM literature, we also adopt a reverse iteration ordering: unlike the standard ADMM algorithm, which counts iterations in ascending order, our algorithm enumerates iterations from a fixed number down to zero. Additional implementation details for PnP-CM are provided in Appendix B.

We also highlight that all the steps of our algorithm are consistent with optimization-based principles. For instance, unlike CM4IR (Garber & Tirer, 2025), which initializes the algorithm with median filtering of $\mathbf{A}^\dagger\mathbf{y}$, our PnP-CM can start from $\mathbf{0}$ initialization, and subsequently follow the algorithm without heuristic modifications and converge in as few as two outer iterations/NFEs.

## 4    EXPERIMENTS

### 4.1    EXPERIMENTAL SETUP AND MODEL IMPLEMENTATION DETAILS

We conduct comprehensive evaluations on the LSUN Bedroom (Yu et al., 2015) and fastMRI (Knoll et al., 2020) datasets, with image sizes of $256 \times 256 \times 3$ and $320 \times 320 \times 2$, respectively. In particular, the New York University fastMRI dataset (Knoll et al., 2020) contains complex-valued images and multi-coil k-space data from fully sampled coronal proton density (PD) and proton density with fat suppression (PD-FS) knee MRI scans, obtained with relevant institutional review board approvals.

For natural images, we use the pre-trained unconditional CM on LSUN Bedroom provided by Song et al. (2023b) without additional retraining. Likewise, we employ the pre-trained unconditional DM on LSUN Bedroom from Dhariwal & Nichol (2021). For evaluation, we sampled 300 images from the LSUN Bedroom validation set and normalized all inputs to the range [0,1].

For MRI, we trained a consistency model from scratch on the fastMRI dataset by first training an EDM model (Karras et al., 2022). To train the EDM and CM models for MRI, we use all 973 volumes (subjects) from the training set of the fastMRI dataset, excluding the first and last five slices of each volume, following the protocol of prior works (Chung & Ye, 2022; Chung et al., 2024). We provide further details for training the EDM and CM networks for MRI in Appendix D.1. For evaluation, we followed the mentioned slice exclusion protocol on 10 volumes from the fastMRI validation set, yielding 228 slices for PD and 240 slices for PD-FS.

### 4.2    EXPERIMENTS ON INVERSE PROBLEMS

**Problem setup.** We evaluate our PnP-CM algorithm on a set of noisy inverse problems. The following tasks are used on the LSUN bedroom dataset: (i) random inpainting with 70% masking,

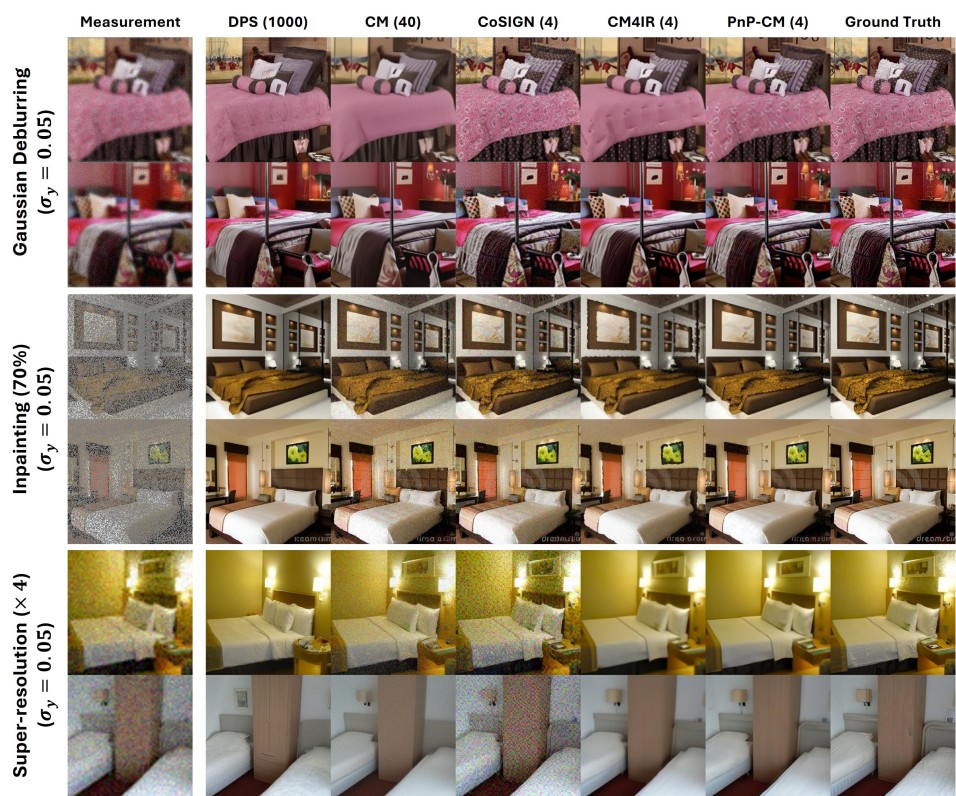

Figure 2: Representative results for Gaussian deblurring, inpainting (70%), and super-resolution (×4). PnP-CM produces sharper and more coherent outputs than prior approaches, avoiding the over-smoothing in DPS and recovering structures that CM-based methods fail to capture, while achieving overall closer similarity to the ground truth.

(ii) bicubic super-resolution with $4\times$ downsampling, and (iii) Gaussian deblurring using a $5 \times 5$ separable kernel with standard deviation $\sigma = 10.0$. All measurements are generated by applying the forward operator to the ground-truth images, with additive Gaussian measurement noise levels of $\sigma_y \in \{0.025, 0.05\}$. For a practical real-world application, multi-coil MRI reconstruction is considered, which is an inherently noisy linear inverse problem. Multi-coil complex-valued MRI datasets were retrospectively undersampled via 1D random subsampling using acceleration factors of $R = 4$ and $R = 8$, retaining 24 and 12 central k-space lines, respectively, as implemented in DDS (Chung et al., 2024).

**Comparison methods.** We conducted extensive comparisons with various methods. For experiments on natural images, we compare our method with DPS (Chung et al., 2023a) as a DM-based approach, and CM (Song et al., 2023b), CoSIGN (Zhao et al., 2024), and CM4IR (Garber & Tirer, 2025) as CM-based solvers. For MRI reconstruction, we benchmark against DPS (Chung et al., 2023a) and DDS (Chung et al., 2024), both of which employ DM priors. All baselines were adopted from their official public implementations. Further implementation details about are in Appendix B.

### 4.3 QUANTITATIVE AND QUALITATIVE RESULTS

Given the differing requirements of natural and medical image tasks, we employ separate evaluation metrics for the two settings. For natural image tasks, reconstruction quality is assessed using peak signal-to-noise-ratio (PSNR) and learned perceptual image patch similarity (LPIPS), while for medical imaging tasks, we use PSNR and structural similarity index (SSIM) as evaluation metrics. Quantitative results are reported as averages over the validation set, while qualitative examples are provided to illustrate the visual fidelity of the reconstructions.

Table 1: Quantitative results for different inverse problems on natural images ($\sigma_y = 0.05$). Best: **BOLD**, second-best: Underlined.

| Method | NFE↓ | SR ×4 | Gaussian Deblurring | Inpainting |
|---|---|---|---|---|
| | | PSNR↑ / LPIPS↓ | PSNR↑ / LPIPS↓ | PSNR↑ / LPIPS↓ |
| DPS (Chung et al., 2023a) | 1000 | 22.19 / 0.335 | 22.88 / 0.286 | 26.38 / 0.231 |
| CM (Song et al., 2023b) | 40 | 23.16 / 0.425 | 24.86 / 0.388 | 24.44 / 0.410 |
| CoSIGN (Zhao et al., 2024) | 2 | 20.33 / 0.569 | 23.32 / 0.416 | 25.56 / 0.339 |
| | 4 | 20.90 / 0.545 | 23.08 / 0.441 | 25.68 / 0.339 |
| CM4IR (Garber & Tirer, 2025) | 2 | 25.00 / 0.387 | 26.90 / 0.320 | 25.84 / 0.304 |
| | 4 | 25.60 / **0.320** | 27.37 / 0.270 | 26.78 / 0.303 |
| PnP-CM (ours) | 2 | 25.32 / 0.348 | 27.43 / 0.263 | 24.17 / 0.373 |
| | 4 | **25.63** / 0.322 | **28.10** / **0.237** | **27.95** / **0.219** |

We implement PnP-CM, as shown in Alg. 1, for multiple ADMM iteration counts ($N \in \{2, 4, 8\}$) and independently fine-tune all coefficients for each task, including the injected noise level and the CM noise level, as well as the momentum coefficients for the primal and dual updates.

**Natural image tasks.** Representative results in Fig. 2 demonstrate that PnP-CM consistently yields sharper and more faithful reconstructions. Compared to DPS, our method avoids oversmoothing while being significantly faster, and it generates coherent structures that CM-based approaches often fail to recover. The reconstructions also preserve high frequency details without introducing the grid-like artifacts seen in competing methods. These visual observations are reflected quantitatively in Tab. 1, where PnP-CM reaches state of the art quality with only a few evaluations. Additional qualitative comparisons are provided in Appendix C.

**MRI Reconstruction.** Representative MRI reconstructions for $R = 8$ are depicted in Fig. 3, showing that our method substantially reduces the blurring artifacts observed in DPS, as well as the structured artifacts present in both DPS and DDS. Tab. 2 summarizes the performance of different approaches on the Coronal PD and PD-FS datasets with acceleration rates of $R = 4$ and $R = 8$. Across all settings, our proposed method with NFE = 4 consistently outperforms both DPS (NFE = 1000) and DDS (NFE = 100) in terms of PSNR and SSIM. Additional visual results, including those at the more modest $R = 4$, are provided in Appendix D.2. We also note that CM4IR fails in this scenario, as the backprojection for its data fidelity amounts to $\mathbf{A}^\dagger \mathbf{y} - \mathbf{x}$ in multi-coil MRI, where $m > n$. In other words, the solution tries to move towards the linear least squares solution, $\mathbf{A}^\dagger \mathbf{y}$, which suffers from substantial aliasing artifacts.

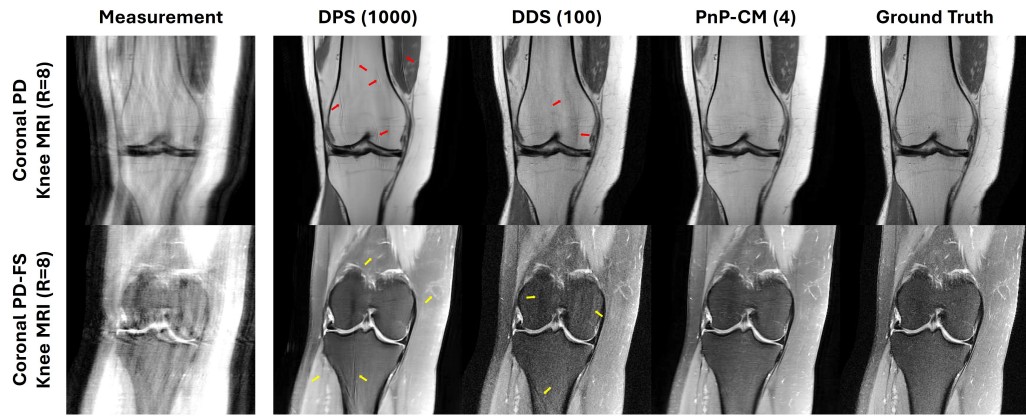

Figure 3: Qualitative comparisons of DPS, DDS, and PnP-CM. **Top:** Coronal PD with $R = 8$. **Bottom:** Coronal PD-FS with $R = 8$. The proposed PnP-CM method effectively reduces artifacts and blurring that are not removed by other methods (red and yellow arrows).

Table 2: Quantitative comparison of reconstruction methods with $R = 4$ and 8 1D random under-sampling for Coronal PD and Coronal PD-FS knee MRI. Best: **BOLD**, second-best: Underlined

| Method | NFE↓ | R | Cor PD | | Cor PD-FS | |
|---|---|---|---|---|---|---|
| | | | PSNR↑ | SSIM↑ | PSNR↑ | SSIM↑ |
| DPS (Chung et al., 2023a) | 1000 | ×4 | $31.21_{\pm4.81}$ | $0.847_{\pm0.097}$ | $\underline{31.17}_{\pm3.25}$ | $\underline{0.778}_{\pm0.079}$ |
| | | ×8 | $29.83_{\pm4.14}$ | $0.823_{\pm0.080}$ | $28.41_{\pm3.89}$ | $\underline{0.731}_{\pm0.092}$ |
| DDS (Chung et al., 2024) | 100 | ×4 | $\underline{31.93}_{\pm4.94}$ | $\underline{0.865}_{\pm0.113}$ | $31.00_{\pm2.78}$ | $0.768_{\pm0.090}$ |
| | | ×8 | $\underline{30.52}_{\pm3.83}$ | $\underline{0.843}_{\pm0.085}$ | $\underline{29.71}_{\pm2.77}$ | $0.721_{\pm0.101}$ |
| PnP-CM (ours) | 4 | ×4 | $\mathbf{32.69}_{\pm3.29}$ | $\mathbf{0.902}_{\pm0.038}$ | $\mathbf{31.82}_{\pm3.02}$ | $\mathbf{0.820}_{\pm0.054}$ |
| | | ×8 | $\mathbf{31.10}_{\pm3.16}$ | $\mathbf{0.872}_{\pm0.047}$ | $\mathbf{30.80}_{\pm2.62}$ | $\mathbf{0.782}_{\pm0.067}$ |

**Ablation Studies.** We conducted an ablation study to disentangle the contributions of momentum and noise injection in our ADMM-based solver. Specifically, we compared four variants: (i) baseline PnP-ADMM, (ii) PnP-ADMM with momentum, (iii) PnP-ADMM with noise injection, and (iv) our PnP-CM combining both momentum and noise injection. Experiments were performed on the LSUN Bedroom dataset for super-resolution, inpainting, and Gaussian deblurring with NFE = 4 and $\sigma_y = 0.05$.

Table 3: Comparison of ADMM strategies and highlighting the effect of momentum and noise injection ($\sigma_y = 0.05$, $N = 4$). Best: **BOLD**, second-best: Underlined.

| Noise Inj. | Momentum | SR ×4 | Gaussian Deblurring | Inpainting |
|---|---|---|---|---|
| | | PSNR↑ / LPIPS↓ | PSNR↑ / LPIPS↓ | PSNR↑ / LPIPS↓ |
| ✗ | ✗ | 20.95 / 0.581 | 27.46 / 0.274 | 17.34 / 0.547 |
| ✓ | ✗ | 20.83/ 0.581 | 27.58 / 0.260 | 17.52 / 0.541 |
| ✗ | ✓ | 25.10 / 0.357 | 28.05 / 0.243 | 27.75 / 0.232 |
| ✓ | ✓ | **25.63 / 0.322** | **28.10 / 0.237** | **27.95 / 0.219** |

The quantitative results given in Tab. 3 consistently demonstrate that both momentum and noise injection improve reconstruction quality. Notably, momentum provides a stronger boost than noise injection across all tasks, establishing ADMM with momentum as the second-best performer after our full method. Our approach, which combines both techniques, achieves the best results overall, highlighting their complementary benefits.

# 5 DISCUSSION

**Limitations.** Due to a lack of public availability of pre-trained CMs, we used LSUN-Bedroom for inverse problems with natural images, as it was the only dataset with an associated CM. However, evaluating the performance of the algorithms on a dataset with more detailed image-level features, such as ImageNet or FFHQ would be desirable to investigate more subtle reconstruction differences.

**Future directions.** In principle, PnP-CM can work with nonlinear forward operators, $\mathcal{A}(\cdot)$. In this case, $\mathbf{f}(x) = ||\mathbf{y} - \mathcal{A}(\mathbf{x})||_2^2$, and Eq. 10 needs to be solved with a gradient descent type method. The algorithmic details of such an implementation will be investigated in future work.

# 6 CONCLUSION

In this work, we introduce PnP-CM, a novel framework that integrates consistency models (CMs) as proximal operators within the PnP optimization paradigm. By combining the efficiency of CMs with the convergence properties of PnP-ADMM methods, PnP-CM offers a significant advance in solving inverse problems. Our method, enhanced with noise injection and momentum, achieves high-quality reconstructions in as few as 2–4 NFEs, demonstrating both precision and speed. We show that PnP-CM excels across a diverse range of inverse problem tasks, including inpainting, super-resolution, deblurring, and MRI reconstruction, with the latter being the *first* application of a CM trained specifically for MRI data. Through extensive evaluations, we demonstrate that PnP-CM surpasses existing state-of-the-art CM-based methods in reconstruction quality, setting a new benchmark for high-fidelity inverse problem solving.

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

## A    PROOF OF THEOREM 1

*Proof.* To analyze the effect of noise injection on the convergence behavior, we compare the combined residuals in the presence and absence of noise, decomposing the residual into its primal and dual components:

$$\Delta_k = \frac{1}{\sqrt{P}}\Big( \underbrace{||\mathbf{z}_{k+1} - \mathbf{z}_k||_2}_{\Delta_{\mathbf{z}_k}} + \underbrace{||\mathbf{x}_{k+1} - \mathbf{x}_k||_2}_{\Delta_{\mathbf{x}_k}} + \underbrace{||\mathbf{u}_{k+1} - \mathbf{u}_k||_2}_{\Delta_{\mathbf{u}_k}} \Big). \tag{14}$$

Since the primal variable $\mathbf{z}$ is updated prior to noise injection, it remains unaffected. We use the superscripts $\mathbf{0}$ and $\boldsymbol{\eta}$ to denote the cases without and with noise injection, respectively.

For the $\mathbf{x}$-update, we obtain

$$\begin{aligned}
\Delta_{\mathbf{x}_k}^{\boldsymbol{\eta}} &= ||\mathbf{x}_{k+1} - \mathbf{x}_k||_2 \\
&= ||f_\theta(\mathbf{z}_{k+1} + \mathbf{u}_k + \boldsymbol{\eta}_k) - \mathbf{x}_k||_2 \\
&= ||f_\theta(\mathbf{z}_{k+1} + \mathbf{u}_k + \boldsymbol{\eta}_k) - f_\theta(\mathbf{z}_{k+1} + \mathbf{u}_k) + f_\theta(\mathbf{z}_{k+1} + \mathbf{u}_k) - \mathbf{x}_k||_2 \\
&\le ||f_\theta(\mathbf{z}_{k+1} + \mathbf{u}_k + \boldsymbol{\eta}_k) - f_\theta(\mathbf{z}_{k+1} + \mathbf{u}_k)||_2 + ||f_\theta(\mathbf{z}_{k+1} + \mathbf{u}_k) - \mathbf{x}_k||_2 \\
&= ||f_\theta(\mathbf{z}_{k+1} + \mathbf{u}_k + \boldsymbol{\eta}_k) - f_\theta(\mathbf{z}_{k+1} + \mathbf{u}_k)||_2 + \Delta_{\mathbf{x}_k}^{\mathbf{0}} \\
&\le L||\boldsymbol{\eta}_k||_2 + \Delta_{\mathbf{x}_k}^{\mathbf{0}}, \tag{15}
\end{aligned}$$

where $L$ is the Lipschitz constant of $f_\theta$.

Similarly, for the $\mathbf{u}$-update we have

$$\begin{aligned}
\Delta_{\mathbf{u}_k}^{\boldsymbol{\eta}} &= ||\mathbf{u}_{k+1} - \mathbf{u}_k||_2 \\
&= ||(\mathbf{u}_k + \mathbf{z}_{k+1} - \mathbf{x}_{k+1}) - \mathbf{u}_k||_2 \\
&= ||\mathbf{z}_{k+1} - \mathbf{x}_{k+1}||_2 \\
&= ||\mathbf{z}_{k+1} - f_\theta(\mathbf{z}_{k+1} + \mathbf{u}_k + \boldsymbol{\eta}_k)||_2 \\
&= ||\mathbf{z}_{k+1} - f_\theta(\mathbf{z}_{k+1} + \mathbf{u}_k) + f_\theta(\mathbf{z}_{k+1} + \mathbf{u}_k) - f_\theta(\mathbf{z}_{k+1} + \mathbf{u}_k + \boldsymbol{\eta}_k)||_2 \\
&\le ||\mathbf{z}_n - f_\theta(\mathbf{z}_{k+1} + \mathbf{u}_k)||_2 + ||f_\theta(\mathbf{z}_{k+1} + \mathbf{u}_k) - f_\theta(\mathbf{z}_{k+1} + \mathbf{u}_k + \boldsymbol{\eta}_k)||_2 \\
&= \Delta_{\mathbf{u}_k}^{\mathbf{0}} + ||f_\theta(\mathbf{z}_{k+1} + \mathbf{u}_k) - f_\theta(\mathbf{z}_{k+1} + \mathbf{u}_k + \boldsymbol{\eta}_k)||_2 \\
&\le \Delta_{\mathbf{u}_k}^{\mathbf{0}} + L||\boldsymbol{\eta}_k||_2 \tag{16}
\end{aligned}$$

Hence, the additional divergence caused by noise injection at iteration $n$ is at most

$$\frac{2L}{\sqrt{P}}||\boldsymbol{\eta}_k||_2. \tag{17}$$

Formally, the total deviation introduced across all iterations is bounded by

$$\sum_k \Delta_k^{\boldsymbol{\eta}} - \sum_k \Delta_k^{\mathbf{0}} \;\le\; \frac{2L}{\sqrt{P}} \sum_k ||\boldsymbol{\eta}_k||_2. \tag{18}$$

The convergence is preserved as long as the cumulative effect of noise injection remains finite, $\sum_{k=1}^{\infty} ||\boldsymbol{\eta}_k||_2 < \infty$, and the noise injection is diminishing. This condition can be enforced by an appropriate noise scheduling strategy, e.g., by decaying the noise magnitude over iterations. Then the additional divergence introduced by noise injection remains bounded, ensuring that the overall convergence behavior of the algorithm is unaffected. $\square$

## B    IMPLEMENTATION DETAILS FOR COMPARISONS AND PROPOSED METHOD

**PnP-CM.**    We implemented our method following Alg. 1. Step 3, which involves solving a linear system, was carried out using singular value decomposition (SVD) for small-scale experiments, consistent with prior works (Song et al., 2023a; Wang et al., 2023; Garber & Tirer, 2025). For large-scale MRI reconstruction, we employed conjugate gradient (CG) method to ensure computational efficiency. We tuned the related hyperparameters empirically, as described in the main text.

**DPS.** We followed the official implementation provided by Chung et al. (2023a). Since the likelihood weight was originally tuned for ImageNet and FFHQ, we re-tuned it for LSUN Bedroom and FastMRI knee datasets, using $\eta = 1.0$ for natural images and $\eta = 0.85$ for medical imaging.

**DDS.** We used the official implementation of Chung et al. (2024) with $\gamma = 1.0$ and $\eta = 0.85$. In contrast to clinical MRI reconstruction, where subsampled k-space measurements already contain inherent noise and no clean reference is available, the original method adds Gaussian noise to clean data. To better reflect the clinical setting, we applied DDS directly on the noisy subsampled k-space measurements. We tuned the CG steps depending on the data SNR.

**CM.** The official public repositories provided by Song et al. (2023b) were followed during the implementation. While the codebase includes iterative functions for inpainting and super resolution, it does not provide one for Gaussian deblurring, which we implemented ourselves. We applied 40 steps for all inverse problems as advised by the authors.

**CoSIGN.** We used the official implementation provided by Zhao et al. (2024). Since they do not provide any ControlNet training checkpoint for inpainting (70%) and Gaussian Deblurring tasks, we trained them from scratch. We used the provided checkpoint for super-resolution ($\times 4$).

**CM4IR.** For CM4IR, we relied on the official implementation and hyperparameter settings provided by Garber & Tirer (2025). Since these configurations were extensively tuned by the authors, we retained them without modification.

## C    ADDITIONAL RESULTS FOR LSUN BEDROOM DATASET

Tab. 4 reports extended quantitative results on the LSUN Bedroom dataset for varying noise levels and sampling steps. PnP-CM maintains consistently strong performance across different inverse tasks, showing robustness to both the number of function evaluations and measurement noise. Increasing the sampling steps generally improves quality as expected, while performance remains competitive even in the few-step regime. In addition to these quantitative comparisons, we also provide further qualitative results against all baseline methods discussed in the main text, illustrating that the visual trends observed there hold across a wider set of examples.

Table 4: Quantitative results on LSUN Bedroom for different measurement noise levels ($\sigma_y$) and varying numbers of NFEs across super resolution ($\times 4$), Gaussian deblurring, and inpainting tasks.

| NFE↓ | $\sigma_y$ | SR $\times 4$ | Gaussian Deblurring | Inpainting |
|---|---|---|---|---|
| | | PSNR↑ / LPIPS↓ | PSNR↑ / LPIPS↓ | PSNR↑ / LPIPS↓ |
| 2 | 0.025 | 25.88 / 0.300 | 28.18 / 0.272 | 24.28 / 0.399 |
| | 0.05 | 25.32 / 0.348 | 27.43 / 0.263 | 24.17 / 0.373 |
| 4 | 0.025 | 26.16 / 0.288 | 29.34 / 0.203 | 28.29 / 0.205 |
| | 0.05 | 25.63 / 0.322 | 28.10 / 0.237 | 27.95 / 0.219 |
| 8 | 0.025 | 26.10 / 0.291 | 30.35 / 0.171 | 29.77 / 0.136 |
| | 0.05 | 25.22 / 0.342 | 28.46 / 0.223 | 29.57 / 0.153 |

## D    ADDITIONAL DETAILS FOR MRI DATASETS

### D.1    TRAINING DETAILS OF DIFFUSION AND CONSISTENCY MODELS FOR MRI

We train an EDM-based diffusion model for MRI (Karras et al., 2022), and for the first time, establish a robust consistency model specifically designed for MRI. Our implementation follows the publicly available code from (Karras et al., 2022; Song et al., 2023b). For U-Net training hyperparameters, we adopt the protocol of CoSIGN (Zhao et al., 2024). Both models are trained on the

fastMRI dataset, as described in Section 4.1. The diffusion model is trained for 1M iterations with a batch size of 4, and is then distilled into a consistency model through an additional 7M iterations using the same batch size.

## D.2 ADDITIONAL QUALITATIVE RESULTS ON MRI RECONSTRUCTION

We provide additional qualitative reconstruction results for coronal PD and PD-FS datasets at acceleration factors $R = 4$ and $R = 8$, with representative examples from each case shown in Fig. 7.

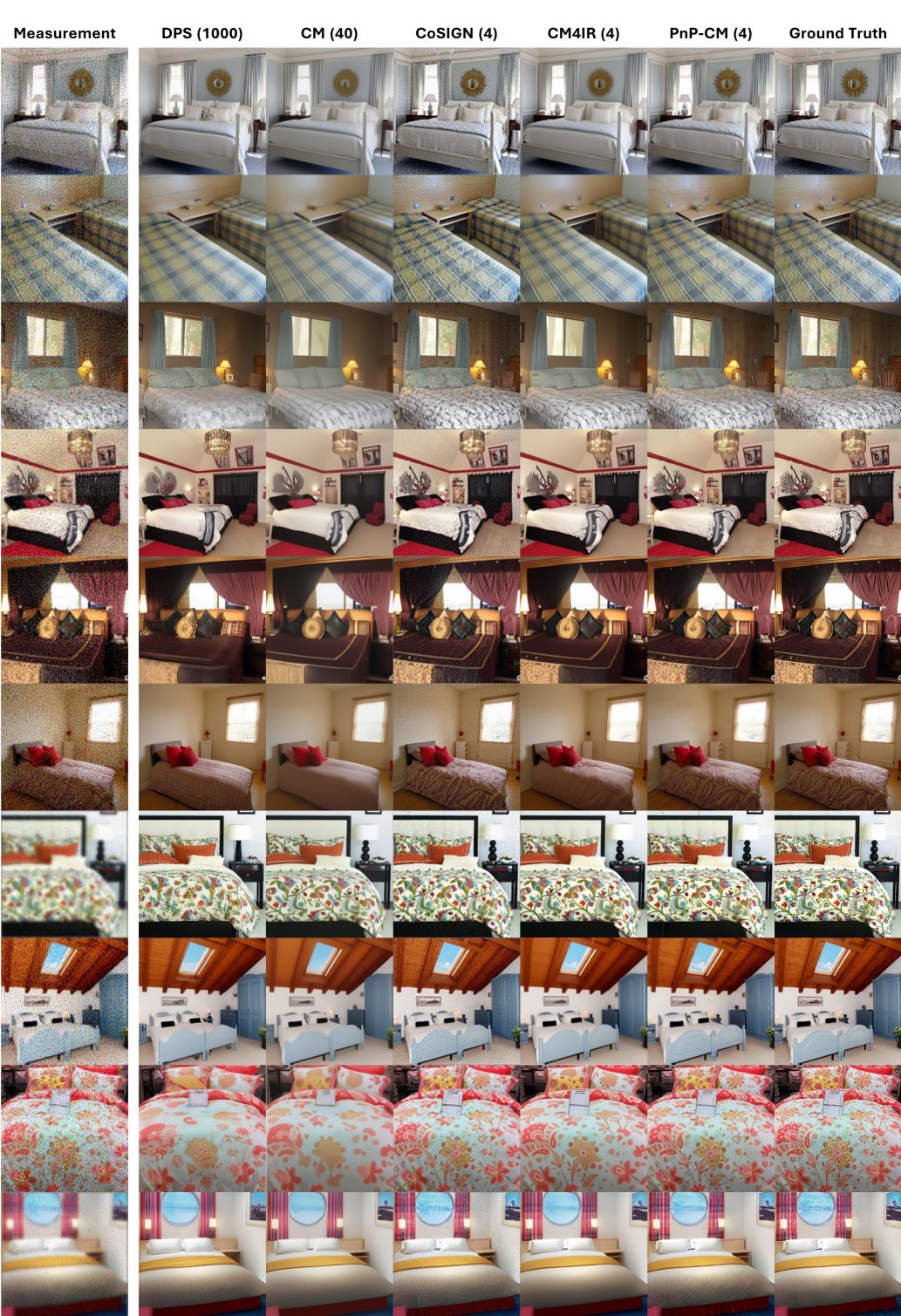

Figure 4: Representative Gaussian deblurring results on LSUN Bedroom with $\sigma_y = 0.05$. Comparisons with all baseline methods show that PnP-CM restores textures more faithfully and avoids oversmoothing.

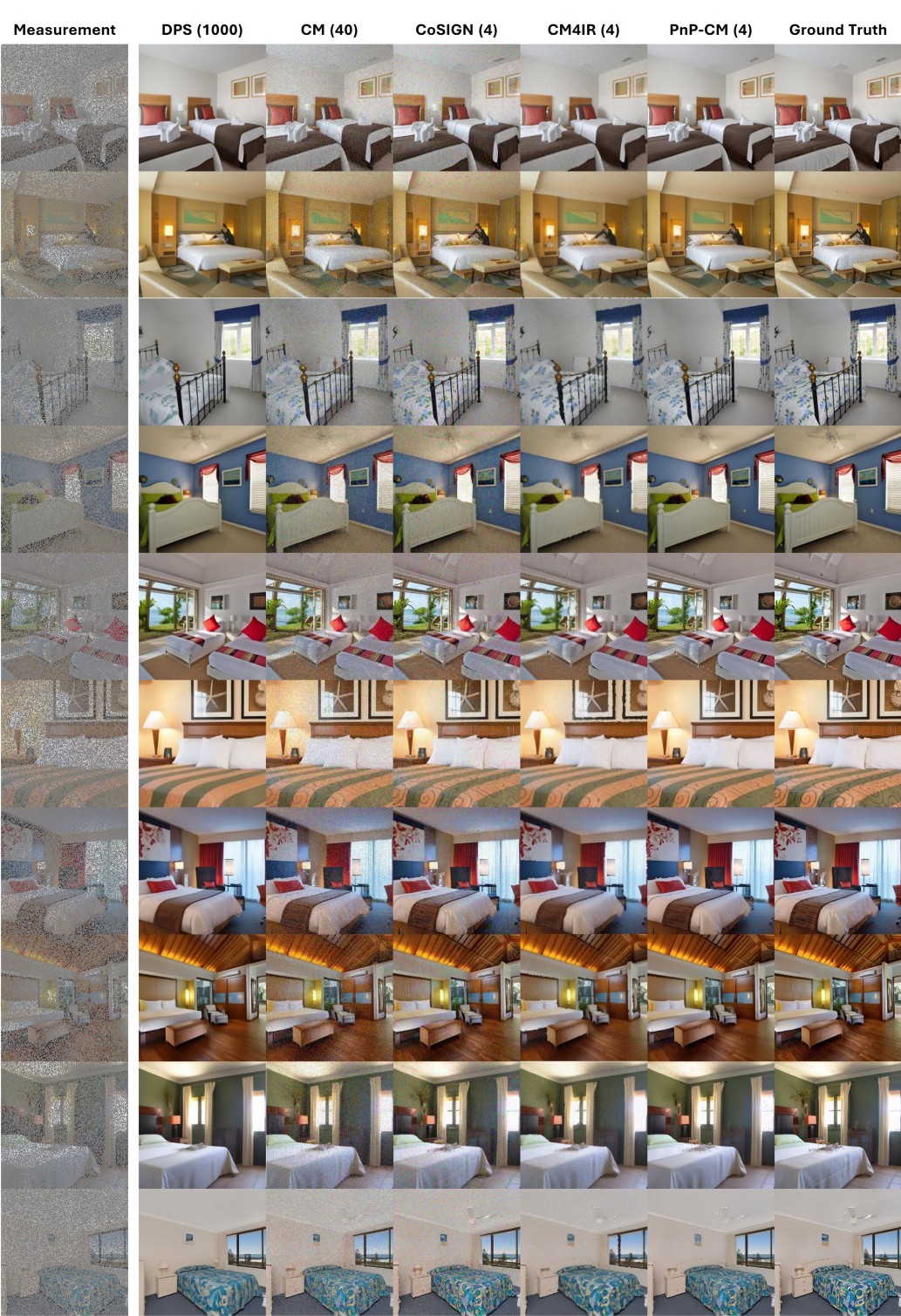

Figure 5: Illustrative inpainting results on LSUN Bedroom with $\sigma_y = 0.05$. Compared to other methods, PnP-CM fills missing regions with coherent structures and yields reconstructions closer to the ground truth.

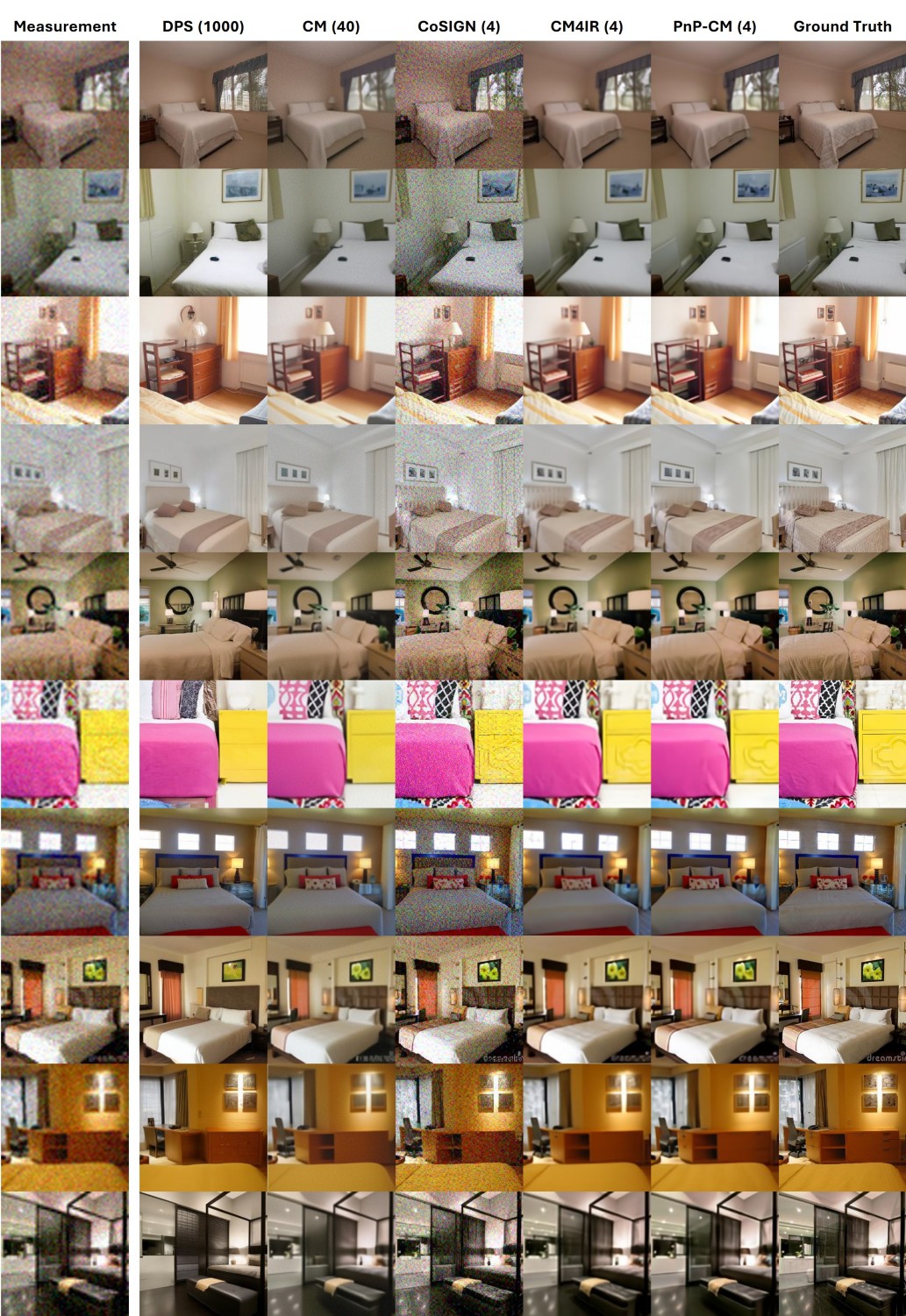

Figure 6: Demonstration of super resolution ($\times 4$) results on LSUN Bedroom with $\sigma_y = 0.05$. Reconstructions are compared against all baseline methods, with PnP-CM producing sharper details and closer resemblance to the ground truth.

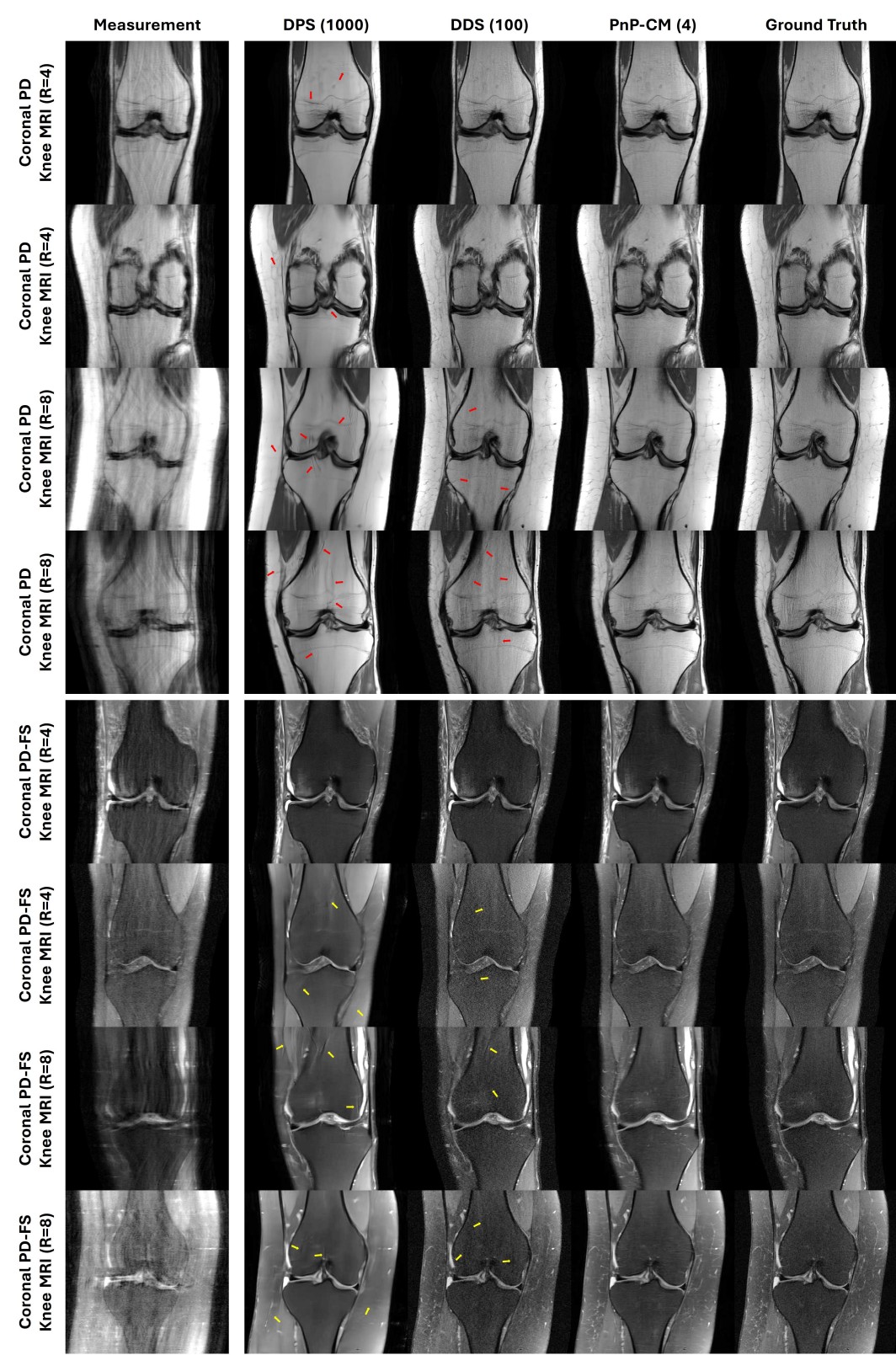

Figure 7: Qualitative comparisons for $R = 4$ and $R = 8$ on the Coronal PD and Coronal PD-FS datasets across different methods. The proposed method, PnP-CM, consistently demonstrates superior performance by effectively reducing artifacts.

