# OpenReview forum: "Consistency Models as Plug-and-Play Priors for Inverse Problems"
_ICLR.cc/2026/Conference — ICLR 2026 Conference Withdrawn Submission_

### Official Review · Reviewer_kjDC · 2025-10-29

**Soundness:** 3
**Presentation:** 2
**Contribution:** 2
**Rating:** 2
**Confidence:** 3

**Summary:**

This paper proposes PnP-CM, a PnP-ADMM framework that employs a Consistency Model (CM) as its denoiser to achieve extremely few-step zero-shot inverse problem solving. To further improve reconstruction dynamics, the authors additionally introduce re-noising before denoising and a Nesterov momentum update within the iterative process. The method demonstrates that high-quality solutions can be obtained with only 2–4 NFEs, which is notably efficient in terms of neural network evaluations.

**Strengths:**

1. The combination of PnP-ADMM with a CM-based denoiser enables inverse problem solving with extremely low NFE. The proposed approach outperforms similar CM-based strategies while avoiding dependency on pseudo-inverses.
2. The method is grounded in a mathematically well-established PnP-ADMM framework, which provides theoretical soundness and stable convergence behavior.
3. The proposed algorithm is simple to implement, as it relies on existing PnP-ADMM and CM pipelines, making it highly reproducible and practical.

**Weaknesses:**

1. The novelty is limited. The work directly adopts CM as a denoiser within PnP-ADMM without introducing a new mechanism tailored for this interaction. Similarly, the use of Nesterov momentum and re-noising steps has been explored in prior PnP and CM literature. Without additional methodological design that specifically leverages CM’s properties, the contribution may be viewed as a straightforward combination of existing techniques.
2. The comparison to baselines is insufficient. DPS is a relatively dated baseline, and many recent few-step diffusion-based inverse problem solvers exist. In particular, comparison with *CM-based few-step solvers* such as LATINO [1] is essential.
3. Important hyperparameter details are missing. How is the sequence of time points $t_n$ scheduled? How are ADMM-related hyperparameters selected? How sensitive is performance to these hyperparameters? These settings likely influence reconstruction quality and should be clearly reported.
4. Due to the few-step nature and characteristics of CMs, there is concern regarding potential **l**oss of fine details / blurring, which may not be adequately captured by metrics like PSNR or LPIPS. For instance, results in Figures 4 and 6 appear noticeably blurred. Additional metrics or qualitative analysis focused on texture/detail preservation would help support the claims.
5. A performance–NFE trade-off analysis is needed. Does increasing NFE improve detail and reduce blur? How does reconstruction quality evolve beyond the extremely small-NFE regime? This would clarify whether the proposed approach is fundamentally limited by its few-step formulation.
6. The reliance on linear inverse formulations prevents the method from being applied in latent space, which is where most modern high-resolution diffusion pipelines operate. This substantially limits the method’s practical applicability. It would be helpful if the authors could discuss whether there exists any conceptual pathway to extend the method to latent-space settings while preserving the required linearity assumptions.
7. Minor issues:
    - L63: “diffusion matching distillation” → “distribution matching distillation”
    - Eq. (11): $u^(k)$ → $u^{(k)}$

[1] Spagnoletti, Alessio, et al. "Latino-pro: Latent consistency inverse solver with prompt optimization." ICCV, 2025.

**Questions:**

1. Under the same algorithmic configuration, what would happen if a standard diffusion model (i.e., denoised estimate) is used instead of a CM? As shown in [1], some methods apply a reverse diffusion process with noise annealing. Without such annealing, would the diffusion model fail to generate solutions?

[1] Zhu, Yuanzhi, et al. "Denoising diffusion models for plug-and-play image restoration."CVPR. 2023.

---

### Official Review · Reviewer_L3sN · 2025-10-31

**Soundness:** 1
**Presentation:** 2
**Contribution:** 2
**Rating:** 4
**Confidence:** 4

**Summary:**

Consistency models (CMs), originally designed to speed up diffusion-based inference, i.e., for generating high-quality samples with low NFEs, have also been adapted recently for inverse problem solving. This paper integrates the use of CMs with a pnp-ADMM styled method, resulting in pnp-CM, particularly meant for MAP estimation using CMs, which can solve downstream inverse problems. Furthermore, noise injection and momentum terms were added for efficiency while retaining the convergence properties of pnp-ADMM. Several image restoration experiments reveal that pnp-CM outperforms existing CM-based solvers in just 2-4 NFEs.

**Strengths:**

The paper proposes a methodology to integrate consistency models with pnp-ADMM framework, further pushing the field of consistency model-based inverse problem solving.

The vanilla integration is further enhanced by noise injection and momentum, resulting in improved convergence of pnp-CM, which solves downstream inverse problems in a few NFEs, particularly crucial in real-time use cases.

Experiments on several image restoration tasks show pnp-CM outperforms existing CM-based methods in as few as 2-4 NFEs. This validates the empirical effectiveness of the method.

The paper is generally well written, well organized, and easy to read.

**Weaknesses:**

Though the paper proposes an insightful contribution, there are concerns regarding the motivation of the proposed method, insufficient comparison with other relevant methods, and why the method only considered easier image restoration inverse problems on only one dataset. How several important hyperparameters are selected is quite unclear. They seem to be quite important for the empirical effectiveness of pnp-CM. Also in this regard, the noise injection and other updates overall look highly engineered in order to achieve high performance using only a few NFEs. Adequately addressing the specific questions below may strengthen the paper.

**Questions:**

Q1. pnp-ADMM is for solving the MAP optimization problem. Previous work, such as [1] and [2], showed that direct optimization of the MAP objective using consistency models is possible. What would be the motivation to use pnp-ADMM-styled optimization solving?

Q2. Most importantly, Lines 229-231, Lines 269-287, and Lines 394-397 show that the method is rather engineered since there are many hyperparameters, such as the penalty parameters and the momentum coefficients, etc, that need to be chosen in Algorithm 1. This is an important concern regarding the empirical effectiveness of the method. Addressing how these hyperparameters are selected and how sensitive pnp-CM is to these hyperparameter choices is quite essential.

Q3. As mentioned in Q1, is the reason for choosing pnp-ADMM styled optimization mainly due to computational efficiency? In that case, does pnp-CM still outperform others if harder inverse problems are considered? Such as a large hole image inpaining task?. From my own experience, random inpainting and Gaussian deblurring are relatively easier tasks. Can pnp-CM still solve these harder tasks in a few NFEs, or would it require comparably more NFEs? Also, I believe a thorough comparison in this regard against works such as [1] and [2] is highly appropriate.

Q4. Also, I believe more datasets are needed to justify the efficiency claims of pnp-CM, since currently all the conclusions were made from experiments on the LSUN dataset and relatively easier inverse problems in the case of natural image restoration. I understand that a new dataset requires training a consistency model; however, the Imagenet64 consistency model is readily available from [4], which the work in [3] also uses, and should be possible to compare against.

Q5. Lastly, training consistency models is quite tricky and requires immense computational resources. So I wonder how the pnp-CM performance would change if one uses a denoiser in place of a consistency model, just like the one-step/multi-step denoiser approximation of the consistency model proposed in [2]. In this case, it may require more NFEs than 2-4, but would pnp-CM still be effective when a diffusion model is used instead? (Line 229 mentions “In this study, we use the CMs as the denoisers in Eq. 13,” without any justification, and note that CMs can not be exact proxies of denoisers as their roles are quite different)


[1] Xu et al. Consistency model is an effective posterior sample approximation for diffusion inverse solvers
[2] Gutha et al. Inverse Problems with Diffusion Models: A MAP Estimation Perspective
[3] Garber et al. Zero-Shot Image Restoration Using Few-Step Guidance of Consistency Models (and Beyond)
[4] Song et al. Consistency Models

---

### Official Review · Reviewer_d222 · 2025-10-31

**Soundness:** 3
**Presentation:** 2
**Contribution:** 2
**Rating:** 4
**Confidence:** 4

**Summary:**

This paper introduces PnP-CM, a plug-and-play ADMM-based inverse problem solver that integrates Consistency Models (CMs) as proximal operators of a learned prior. The method is further accelerated using noise injection and Nesterov momentum, achieving high-quality reconstructions in as few as 2–4 neural function evaluations (NFEs). The authors evaluate PnP-CM on a range of linear inverse problems including inpainting, super-resolution, deblurring, and MRI reconstruction—the latter using what they claim is the first CM trained specifically for MRI data. Experimental results show that PnP-CM outperforms existing CM-based and diffusion-based baselines in both quality and efficiency.

**Strengths:**

PnP-CM achieves competitive reconstruction quality in only 2–4 NFEs, significantly faster than existing diffusion-based methods (e.g., DPS, DDS) and other CM-based solvers (e.g., CoSIGN, CM4IR).

**Weaknesses:**

- The core idea of using CMs as plug-and-play priors has been explored in prior work, notably CM4IR (Garber & Tirer, CVPR 2025). The modifications proposed here—e.g., solving the data fidelity term via a proximal operator and using ADMM instead of PGD—are incremental.

- The success of Nesterov momentum in this setting is surprising given its known instability in PnP methods. The authors do not clarify whether momentum is generally applicable or only effective with CMs. An ablation comparing momentum in CM vs. standard diffusion priors is missing.

- The claim that training a CM for MRI is a contribution is overstated. The writing occasionally lacks maturity, e.g., in the abstract and motivation sections.

- While PnP-CM outperforms CM4IR, the reasons for CM4IR’s strong performance (without the momentum term used in the proposed method) are not analyzed. It is unclear whether CM4IR implicitly uses a momentum-like mechanism or benefits from its pseudo-inverse guidance.

- The analysis of hyperparameter tuning is missing. The performance of PnP-like method is highly sensitive to the hyperparameters [1]. It's thereby necessary to carefully investigate the impact of hyperparameters especially the newly added momentum term coefficients.

[1] TFPnP: Tuning-free Plug-and-Play Proximal Algorithms with Applications to Inverse Imaging Problems, JMLR 2022

**Questions:**

- Is the effectiveness of Nesterov momentum specific to CMs, or would it also accelerate standard diffusion-based PnP methods? Could the authors test momentum in a baseline DM-based PnP-ADMM setup?

- CM4IR performs well despite not using momentum. Does it employ an acceleration mechanism similar to momentum? Could the authors analyze the optimization dynamics of CM4IR to better contextualize their own contributions?

---

### Official Review · Reviewer_d8pH · 2025-10-31

**Soundness:** 3
**Presentation:** 3
**Contribution:** 3
**Rating:** 6
**Confidence:** 3

**Summary:**

This paper introduces a method for using consistency models (CM) in plug-and-play (PnP) inverse problems solvers by formulating an ADMM scheme with a consistency model for the proximal operator. In addition to this, the authors point out the need for momentum and noise in their updates. Importantly, the authors show that their method not only outperforms existing CM models for reconstruction in outright image quality but also that their method can perform well in few NFEs.

**Strengths:**

Overall the approach proposed by the authors is interesting and clearly shows visual and numerical improvements in image quality compared to other techniques while using fewer NFEs which is extremely import in many computational imaging settings. The ability for their method to generalize to different non-linear inverse problem settings in the future is also a plus over existing methods. The experiments are well done comparing to other SOTA techniques and clearly prove their methods performance gain.

**Weaknesses:**

There are already many existing techniques already utilizing DMs and CMs for solving inverse problems which does impact the novelty. The authors state that these existing methods lack generalization to other non-linear settings which can be a drawback. It would be great to then show their proposed method's performance on a non-linear problem like phase retrieval. The authors did acknowledge this in the paper and leave it for future work so I dont think this is a major weakness but if part of the argument is that existing methods cant readily be used in other types of inverse problems then it would be good to see the proposed method's performance there. Although showing the NFEs is good to see it would also be great if the authors provided run times for the various methods.

**Questions:**

1. what are the run times for the various methods?
2. It would be helpful to see PSNR SSIM etc. metrics in the reconstruction figures (ex. Figs. 2,3)

---

### Note · Authors · 2025-11-14

I have read and agree with the venue's withdrawal policy on behalf of myself and my co-authors.